# Tachycardiomyopathy in Patients without Underlying Structural Heart Disease

**DOI:** 10.3390/jcm8091411

**Published:** 2019-09-08

**Authors:** Giulia Stronati, Federico Guerra, Alessia Urbinati, Giuseppe Ciliberti, Laura Cipolletta, Alessandro Capucci

**Affiliations:** Cardiology and Arrhythmology Clinic, Marche Polytechnic University, University Hospital “Umberto I–Lancisi-Salesi”, 60126 Ancona, Italy

**Keywords:** arrhythmias, atrial fibrillation, cardiomyopathy, heart failure, supraventricular arrhythmia, systolic dysfunction, tachycardiomyopathy, ventricular arrhythmia

## Abstract

Tachycardiomyopathy (TCM) is an underestimated cause of reversible left ventricle dysfunction. The aim of this study was to identify the predictors of recurrence and incidence of major cardiovascular events in TCM patients without underlying structural heart disease (pure TCM). The prospective, observational study enrolled all consecutive pure TCM patients. The diagnosis was suspected in patients admitted for heart failure (HF) with a reduced ejection fraction and concomitant persistent arrhythmia. Pure TCM was confirmed after the clinical and echocardiographic recovery during follow-up. From 107 pure TCM patients (9% of all HF admission, the median follow-up 22.6 months), 17 recurred, 51 were hospitalized for cardiovascular reasons, two suffered from thromboembolic events and one died. The diagnosis of obstructive sleep apnoea syndrome (OSAS, hazard ratio (HR) 5.44), brain natriuretic peptide on admission (HR 1.01 for each pg/mL) and the heart rate at discharge (HR 1.05 for each bpm) were all independent predictors of TCM recurrence. The left ventricular ejection fraction at discharge (HR 0.96 for each%) and the heart rate at discharge (HR 1.02 for each bpm) resulted as independent predictors of cardiovascular-related hospitalization. Pure TCM is more common than previously thought and associated with a good long-term survival but recurrences and hospitalizations are frequent. Reversing OSAS and controlling the heart rate could prevent TCM-related complications.

## 1. Introduction

Tachycardiomyopathy (TCM) is an important cause of dysfunction of the left ventricle [1]. It is defined as an arrhythmia induced cardiomyopathy in which the impairment of the left ventricle is secondary to rapid and/or asynchronous, irregular myocardial contraction and is partially or completely reversible after treatment of the triggering arrhythmia [2]. Both atrial and ventricular arrhythmias, as well as the premature atrial or ventricular complexes have been noted to cause TCM [1] and no specific heart rate cut-off at which the condition develops has been identified [3].

The first descriptions of TCM were collected by Phillips and Levine in 1949. In their milestone paper, they hypothesized that patients with long-lasting atrial fibrillation could develop heart failure without any other evidence of structural heart disease, and such heart failure could completely disappear after the restoration of the sinus rhythm [4]. TCM is nowadays classified as a non-familial cause of dilated cardiomyopathy, although doubts have been cast on the inclusion of such a disease among those conditions directly affecting the structure and/or function of the heart [5].

TCM is estimated to be under-recognized [3] and the incidence and prevalence of the condition are currently unknown. The mechanisms of TCM and pathways responsible in individual patients are not fully understood [2,6], however it is hypothesised that subclinical ischaemia, abnormalities in energy metabolism, and an overload of calcium and oxidative stress play a role in the pathogenesis of the condition [1]. To this day, two categories of the disease have been described: Arrhythmia-induced TCM, where the arrhythmia is the sole reason for the dysfunction, and arrhythmia-mediated TCM, where the arrhythmia can exacerbate or worsen heart failure (HF) or an underlying heart disease [1]. The former can also be referred to as “pure” TCM and the latter as “impure” TCM [7,8].

The diagnosis of TCM is retrospective and based on the evidence of recovery after appropriate treatment. In fact, although, an arrhythmia is present with a concomitant left ventricular ejection fraction (LVEF) impairment, a cause-effect relationship is not always ascertainable [2]. There is very little data regarding the recurrences and adverse events in patients with TCM in the current available literature.

The aim of this study was to identify the possible predictors of recurrence and long-term morbidity and mortality of pure TCM.

## 2. Materials and Methods

### 2.1. Study Population

This is a prospective, observational study taking into account all patients admitted for acute HF with reduced ejection fraction from January 2012 to the end June 2018 in the Cardiology and Arrhythmology Clinic of the University Hospital “Ospedali Riuniti” of Ancona, Italy, and presenting with evidence of atrial or ventricular arrhythmias on admission.

The potential triggering arrhythmias considered were: Atrial fibrillation (AF), atrial flutter, supraventricular tachycardia, ventricular tachycardia and premature atrial and ventricular complexes. More specifically, this study considered significant > 20000 premature ventricular complexes per day, according to the current available literature [8].

The selection process is detailed in Figure 1 and consisted of two main phases. The first phase started with the hospitalization and aimed at detecting all potential patients with pure TCM. In order to assess the real weight of TCM in clinical practice, the patients with arrhythmia-mediated (impure) TCM were excluded, ruling out all structural or functional heart diseases. Ischemic heart disease was defined as a previous history of revascularization, or evidence of significant coronary obstruction at coronary angiography performed during hospitalization. The patients with non-significant coronary atherosclerosis and no clinical instrumental signs of ischemia were not excluded. Valvular heart disease was defined as a previous history of aortic or mitral replacement or the repair, evidence of severe aortic or mitral regurgitation, severe aortic stenosis, or moderate or severe mitral stenosis. Congenital heart diseases, cardiac amyloidosis, hypertrophic cardiomyopathy, myocarditis, non-compaction cardiomyopathy, post-partum cardiomyopathy, arrhythmogenic right ventricular dysplasia, Fabry’s disease and alcoholic cardiomyopathy were defined according to the current standards. The non-invasive and invasive diagnostic procedures, such as coronary angiography or cardiac magnetic resonance, were performed according to clinical suspicion in all eligible patients. All patients were treated for HF and underwent rhythm or rate control strategies according to the current guidelines [9,10,11,12]. After discharge, the second phase of the selection process started and aimed at confirming pure TCM out of all potential patients (Figure 1). All the patients with suspected TCM were followed-up in the heart failure outpatient clinic at one month and three months after discharge, and twice a year from then on. The patients presenting an improvement of at least one New York Heart Association (NYHA) class and the recovery of at least five points of the left ventricular ejection fraction (LVEF) during the follow-up were diagnosed with arrhythmia-induced (pure) TCM and included in the analysis (Figure 1) [1,13].

The study was conducted according to institutional guidelines, national legal requirements, European standards and the revised Declaration of Helsinki. Being an observational study, a formal approval of the ethics committee was not sought. All patients provided prior written informed consent for anonymous collection and publication of their clinical data. The present report complies with the STROBE initiative (Appendix A) [14].

### 2.2. Endpoints

The primary endpoint was the recurrence of TCM, defined as a new episode of acute HF with reduced ejection fraction along with the evidence of atrial or ventricular arrhythmia, occurring after complete clinical (no HF symptoms) and echocardiographic (LVEF ≥ 50%) recovery of the original episode.

The secondary endpoints were: Death from all causes, major adverse cardiovascular events (defined as non-fatal stroke, non-fatal myocardial infarction and cardiovascular death), and cardiovascular hospitalizations. Cardiovascular hospitalizations were defined as any hospitalization longer than 12 h for one or more of the following reasons: Acute coronary syndrome, unstable angina, HF, atrial or ventricular arrhythmia, valvular heart disease, infective endocarditis, myocarditis, pericarditis, aortic disease, pulmonary embolism, stroke/transient ischemic attack, syncope, cardiovascular-related elective and urgent procedures and complications of such procedures.

### 2.3. Data Collection

Two expert physicians were responsible for the prospective data collecting regarding the patients’ demographics, risk factors, medical history and treatment. Continuous 12-lead ECG monitoring (Mortara Rangoni, Arezzo, Italy) was used to assess the heart rate during hospitalization and underlying arrhythmias. The blood samples for brain natriuretic peptide (BNP) and troponin I were collected on admission and at discharge. Echocardiographic examinations were performed with a monoplane ultrasound probe 4 MHz (M4S) of Vivid 7 Pro (GE Medical Systems, Milwaukee). The digital loops were captured, recording at least three consecutive beats, and analysed off-line using a dedicated software (EchoPAC 13.0; GE Medical Systems, Milwaukee) according to the most recent recommendations. A complete echocardiogram was performed on admission, at discharge, at a 3-month follow-up. Serial echocardiograms were then performed at least every 6 months until complete recovery of LVEF. The echocardiographic loops were obtained with the patient supine and in the left lateral decubitus at the end of a normal breath, minimizing the depth in order to optimize the frame rate (40–80 fps).s LVEF was calculated by the Simpson biplane method. All echo exams were reviewed by two authors (G.S. and F.G.), who were responsible for the off-line analysis and collected all measurements blinded to the recurrence or other clinical endpoints. The inter-operator coefficient of variations for LVEF was 3.2% and the intra-operator coefficient of variation was 2.4%.

To allow for the comparability of drug regimens across the patients taking many different medications, a treatment intensity score (TIS) was calculated. As previously reported, [15] the recorded daily dose taken by the patient was divided by the maximum recommended daily dose to obtain a proportional dose for that medication, called intensity. The maximum recommended daily doses were set by the European and American guidelines [9,10,11,12].

### 2.4. Statistical Analysis

All continuous variables were checked for normality through the Kolmogorov-Smirnov test. The normally-distributed variables were described by the mean and standard deviation and compared by analysis of variance. The not-normally-distributed variables were described as the median and 1st–3rd IQR and compared by non-parametric tests. The categorical variables were described as the absolute and relative values, and compared by chi-square test or Fisher exact test, as appropriate.

The Kaplan-Meier analysis was used in order to assess the time free from primary and secondary endpoints. The association between the individual variables and the risk of TCM recurrence and cardiovascular hospitalization was investigated by using univariate Cox proportional hazards models. The variables that showed an association with each endpoint with a significance level < 0.1 on univariate analyses were entered into the multivariable Cox proportional hazards model. The independent risk factors for each endpoint were then presented as hazard ratios (HRs) and 95% confidence intervals (CIs).

The linearity assumption of the relationship between the independent continuous risk factors and the outcome of interest was represented using restricted cubic splines with three knots located to the 10th, 50th, and 90th percentiles according to the Harrell rule, and assessed by the Wald test for linearity.

SPSS 22.0 for Windows (SPSS Inc., Chicago, IL, USA) and R (R Foundation for Statistical Computing, Vienna, Austria) were used for statistical analysis. The values of *p* < 0.05 (two-tailed) were considered as statistically significant.

## 3. Results

The population included 107 patients (68 males, mean age 66.7 ± 14.5 years). The patients’ characteristics are summarized in Table 1. The median follow-up was 22.6 months (1st–3rd quartile 10.0–40.0 months). The median hospitalization time (i.e., the first phase of the selection process) was 7 days (1st–3rd quartile 4–11 days). The median time to TCM diagnosis confirmation (i.e., the second phase of the selection process) was 72 days (1st–3rd quartile 48–130 days). Eighty-three patients (77.6%) were diagnosed with atrial fibrillation (AF) as underlying arrhythmia, and 16 (15.0%) with atrial flutter. Other triggering arrhythmias included non-sustained ventricular tachycardia (4, 3.7%), paroxysmal supraventricular tachycardia (1, 0.9%) and premature ventricular contractions (PVCs) (3, 2.8%).

During the follow-up, 17 patients experienced at least one recurrence (15.8% of all patients) and 51 were hospitalized for cardiovascular reasons (47.7%). Among the major adverse cardiovascular events, two patients suffered from thromboembolic events (1.8%) and one died from cardiovascular causes (0.9%). No non-fatal myocardial infarctions were reported. The annual incidence of recurrence was 8.4% per year, 0.9% per year for thromboembolic events and 0.4% per year for cardiovascular mortality.

The only death occurred in a 58-year old male, one year and three months after recovery of both NYHA class and LVEF. The patient had no evidence of progression to any kind of structural heart disease and died suddenly after an out-of-hospital cardiac arrest due to idiopathic ventricular fibrillation. One transient ischemic attack and one non-fatal stroke occurred in two different patients after 3 and 832 days, respectively. The treatment strategies at discharge are described in Table 2.

### 3.1. Tachycardiomyopathy Recurrences

Out of the 17 patients experiencing recurrences, seven had multiple recurrences, with six patients experiencing two recurrences and one experiencing four. The arrhythmic disorder underlying TCM recurrences was AF in 15 cases (88%) and atrial flutter in two cases (12%).

The majority of recurrences occurred between the fifth and the sixth year after the first diagnosis as seen in Figure 2.

The multivariate Cox regression analysis showed that presence of obstructive sleep apnoea syndrome (OSAS), BNP on admission and the heart rate at discharge were all independent predictors of TCM recurrence (Table 3).

The univariate model included: The male gender, age, body mass index, hypertension, diabetes, dyslipidaemia, chronic kidney disease, chronic obstructive pulmonary disease, OSAS, hyperthyroidism, hypothyroidism, type of arrhythmia, NYHA class on admission and at discharge, heart rate on admission and at discharge, BNP on admission and at discharge, Troponin I on admission and at discharge, LVEF on admission and at discharge, iLAV, rate or rhythm control (dummy variable) and pharmacological treatment at discharge (with each drug class from Table 2 considered as a separate variable). The complete model is detailed in Appendix A.

BNP: brain natriuretic peptide; CI: confidence interval; iLAV: indexed left atrial volume; LVEF: left ventricular ejection fraction; NYHA: New York Heart Association; OSAS: obstructive sleep apnoea.

According to the spline curves, the heart rate at discharge and the risk or TCM recurrence had a linear association (Appendix A, s*p* for linearity < 0.001). The mean values for both the heart rate and LVEF throughout the index event, the follow-up and at the time of recurrence are shown in Figure 3a,b. Consistently with other statistical models, the heart rate at discharge and during follow-up is significantly higher in patients experiencing a recurrence when compared with the patients with no recurrence. Furthermore, all patients showed a LVEF ≥50% at a 1-year follow-up.

Comparing TIS for each drug class considered in Table 2 showed no differences between patients with and without recurrences (all *p* > 0.05).

From the 17 patients experiencing recurrences, 13 were under a rhythm-control strategy and four were under a rate-control strategy. Twelve patients underwent catheter ablation of AF and two underwent ablation of atrial flutter. The patients undergoing catheter ablation of atrial flutter experienced no further TCM recurrences, while five patients presented a second TCM recurrence after the procedure. Three patients with AF refused consent to catheter ablation and were therefore shifted to a rate-control strategy, with one patient experiencing no further recurrences, one experiencing a second recurrence and another patient experiencing three more recurrences over the follow-up.

### 3.2. Cardiovascular Hospitalizations

From the 51 patients hospitalized for cardiovascular reasons during the follow-up, 15 were hospitalized more than once.

More than 40% of all cardiovascular-related hospitalizations occurred within the first year after the first diagnosis (Figure 4).

EF at discharge and the heart rate at discharge resulted as independent predictors of cardiovascular-related hospitalization according to the multivariate Cox regression model (Table 4 and Appendix A).

The univariate model included: The male gender, age, body mass index, hypertension, diabetes, dyslipidaemia, chronic kidney disease, chronic obstructive pulmonary disease, OSAS, hyperthyroidism, hypothyroidism, type of arrhythmia, NYHA class on admission and at discharge, heart rate on admission and at discharge, BNP on admission and at discharge, Troponin I on admission and at discharge, LVEF on admission and at discharge, iLAV, rate or rhythm control (dummy variable) and pharmacological treatment at discharge (with each drug class from Table 2 considered as a separate variable). The complete model is detailed in Appendix A.

BNP: brain natriuretic peptide; CI: confidence interval; iLAV: indexed left atrial volume; LVEF: left ventricular ejection fraction; NYHA: New York Heart Association; OSAS: obstructive sleep apnoea.

Comparing TIS for each drug class considered in Table 2 showed no differences between the patients with and without cardiovascular hospitalizations (all *p* > 0.05).

## 4. Discussion

The main message of the present study is that, while TCM is associated with an overall good prognosis, TCM patients do recur over a long-time follow-up.

This represents an outstanding difference between HF patients and pure TCM patients as the former is known to be a progressive, worsening condition commonly culminating in the patient’s exitus. Although under-recognized, this study shows that almost 10% of all hospitalizations for acute HF meet the diagnostic criteria for TCM. Therefore, early recognition of the possible triggering arrhythmia is of paramount importance as it can lead to treatment strategies which can favour patient recovery. A clinical suspicion of TCM should arise in all patients presenting with new and quickly worsening symptoms of HF, a low overall cardiovascular risk profile and the recent evidence of high-rate arrhythmia. In these cases, a prompt reduction of the heart rate (either through rate control drugs or restoration of sinus rhythm) should be performed as soon as possible, and possibly even while the diagnostic workup for the exclusion of structural heart disease is still in progress. A cardioversion attempt should be made (when feasible) in order to prevent further deterioration of the systolic function and catheter ablation should be taken into serious consideration [10]. Moreover, in these patients, a sleep study and polysomnography should be performed as soon as possible, even during the hospital stay, as potentially able to unmask OSAS.

The rate of TCM recurrence is higher between the fifth and the sixth year after diagnosis. It can only be speculated that this could be due to the natural progressive reduction of the patients’ adherence to treatment over time.

Our multivariate analysis found three major independent predictors of TCM recurrence. The most important was a concomitant diagnosis of OSAS, which increased the risk of recurrence 5-fold. It was hypothesized that this could be related to the fact that OSAS can alter the physiological parasympathetic modulation of the heart during sleep leading to sympathetic excitation and favouring ventricular and atrial ectopic beats [16,17]. Moreover, OSAS has been described to be an independent risk factor for AF and has been shown to decrease the success rate of antiarrhythmic drugs, electrical cardioversion and catheter ablation [18], potentially leading to TCM recurrence. Despite the lack of information regarding the actual adherence to non-invasive ventilation, it is noted that half of the patients with OSAS and TCM recurrence were not treated with continuous positive airway pressure at all. Therefore, it appears important to educate patients affected by OSAS on the importance of non-invasive ventilation while offering the best treatment strategy in order to improve long-term compliance.

Another striking result that warrants discussion is that the heart rate at discharge is associated with an increased risk of TCM recurrence. More precisely, for each increased beat per minute, the risk of recurrence increases by 5%. Moreover, this association proved to be linear, at least within the ranges of the heart rate seen in our population, and holds true independently of the rhythm at discharge, the treatment strategy and the class of medications used. To make an example, a patient with a lenient rate control strategy (110 bpm) has a 2.5-fold risk of TCM recurrence when compared to the same patient undergoing a strict rate control strategy (80 bpm). This is in contrast to the known evidence that both the heart rate targets are considered similarly effective in preventing adverse events in patients with AF [19]. The reasons for such a striking difference can be found in the different pathophysiological mechanisms. In the RACE II trial, AF patients with severe HF or with recent decompensation were excluded, thus leaving only patients without HF or with stable mild symptoms for at least three months [20]. In this setting, it has already been demonstrated that the actual benefit from the heart rate reduction and sinus rhythm restoration could be counterbalanced by the increased likelihood of adverse effects due to anti-arrhythmic drugs [21] and, therefore, pushing too hard on heart rate reduction could produce no further clinical benefits. On the other hand, it is well known that the heart rate is a risk factor in patients with HF, even when the sinus rhythm is present. Dysfunctional myocardium is energetically depleted and myocardial exerted force is negatively associated with the rate of contraction [22]. In an HF setting, such as the one of TCM occurrence, reducing the heart rate improves contractility, extends coronary diastolic filling time, reduces energy expenditure and improves cardiac output [23]. Moreover, the benefits of a reduced heart rate are consistent over the years, due to the positive modifications of the extracellular matrix and myocytes properties [24], resulting in a reduced risk of cardiovascular events and HF recurrences over a long follow-up. Regarding BNP, a small study already demonstrated that a NT-proBNP drop after four weeks was able to identify TCM with a sensitivity of 84% and a specificity of 95% [25]. In our population, this study found that BNP during the acute phase is an independent predictor of recurrences. This adds evidence to the notion that the patients with pure TCM may benefit from a continuation of HF treatment even after normalization of LVEF in order to prevent recurrences, even if the usefulness, duration and safety of HF treatment in TCM still represent an unexplored grey area.

Although recent reviews and small case series [1,26,27] have hypothesized the TCM recurrences may be characterized by a more severe onset of the condition, our prospective study on a larger population, actually showed that the recurrences are characterized by a higher LVEF and a reduced heart rate. This could be related to the rate-control strategy and to the continuation of HF treatment after discharge. In our population, 15 out of 17 patients had AF as the trigger of TCM recurrence. Therefore, it is feasible to hypothesize that the progressive nature of AF could contribute to the risk of TCM recurrence. Furthermore, when compared to other supraventricular arrhythmias, such as atrial flutter or atrioventricular node re-entry tachycardia, currently available pharmacological and non-pharmacological rhythm control strategies for AF are surely less effective in obtaining an optimal and long-lasting restoration of the sinus rhythm [10].

Regarding major clinical events, there were a few and potentially unrelated to the combination between HF and tachyarrhythmia. Finally, in terms of cardiovascular related hospitalizations, almost half of this study’s population was re-hospitalized, even though by definition, none had structural heart disease. Most hospitalizations occurred during the first year after the event and were related to rhythm control procedures, such as elective cardioversions and catheter ablations. Moreover, 16 hospitalizations were due to the recurrence of TCM. The heart rate at discharge confirmed its predicting value along with the LVEF at discharge. Similar to the fact that heart rate reduction has been demonstrated to be beneficial in HF [23], our data confirm the role of the rate control in the pathophysiology of this peculiar, reversible form of systolic dysfunction. This strengthens the message that, in pure TCM, the lower the heart rate at discharge, the better the long-term prognosis.

### Limitationss

This paper shares all the limitations characterizing all prospective observational studies. In addition, this study’s population was relatively small, and the low sample size made subgroup analyses unfeasible. Nonetheless, current available literature relies on case series and, to our knowledge, this is the largest dataset on pure TCM taken into account so far.

The cut-off used to define pure TCM (improvement of at least one NYHA class and > 5% EF) could seem rather small, but unfortunately, there is no consensus on any cut-off for TCM. Although surely arbitrary, the authors chose this cut-off because it was thought that, after ruling out all causes of structural heart disease, a patient undergoing a clinical and echographic improvement could be considered as having TCM, being the arrhythmia the only remaining and plausible cause of his/her condition. A higher cut-off, as the one proposed by Jeong and colleagues [13], could have ruled out many TCM that just had not time to recover completely because of arrhythmic recurrence, without offering alternative explanations behind the first decompensation. Moreover, according to Table 3, all the patients reached a LVEF of 50% or more after one year, making the authors quite confident that the population was correctly selected.

Furthermore, HF treatment could be considered as a potential confounder in the association between the heart rate and EF improvement/worsening. However, the heart rate was a predictor of the recurrence independently of any kind of pharmacological treatment at discharge (Table 3). As the criteria for TCM recurrence are the same as for the first event, it can be hypothesized that, in the patients, the heart rate is what matters and the association with LVEF worsening could be considered as independent of HF treatment. Of course, subsequent modification or intensification of the HF therapy over time could have modified the strength of such an association, but there is no means to assess that as it would be a daunting task to properly include all treatment changes in the statistical models.

## 5. Conclusions

In conclusion, TCM is an under-diagnosed entity, affecting nearly one out of ten patients admitted for HF. Pure TCM (i.e., without underlying structural heart disease) is associated with a good long-term survival. Nonetheless, recurrences are frequent and can occur after many years. The treatment aimed at reversing OSAS and lowering the heart rate after the acute event could prevent these recurrences and their related hospitalizations.

## Figures and Tables

**Figure 1 jcm-08-01411-f001:**
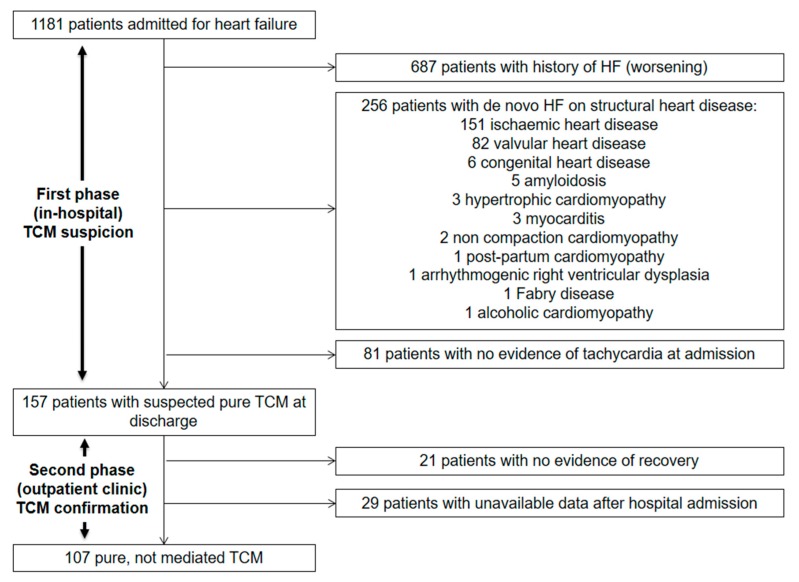
Selection process.

**Figure 2 jcm-08-01411-f002:**
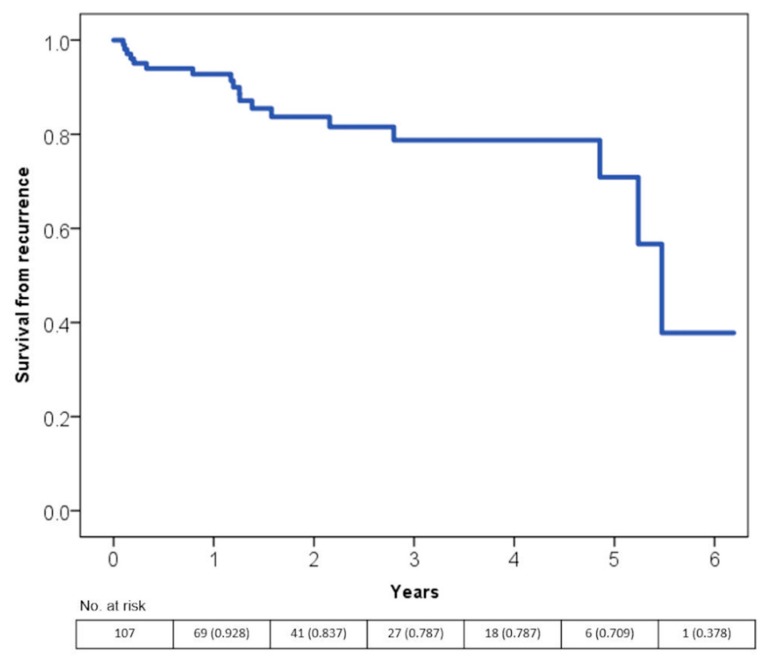
Time free from tachycardiomyopathy recurrence according to the Kaplan-Meier curves.s.

**Figure 3 jcm-08-01411-f003:**
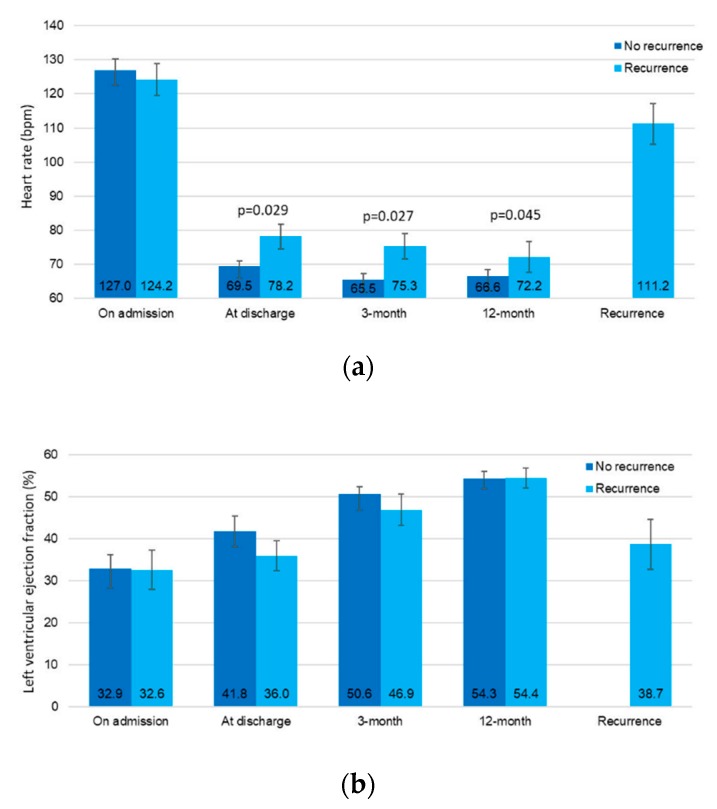
The mean values of the heart rate (**a**) and let ventricular ejection fraction (**b**) during follow-up, according to the presence or absence of future recurrences.

**Figure 4 jcm-08-01411-f004:**
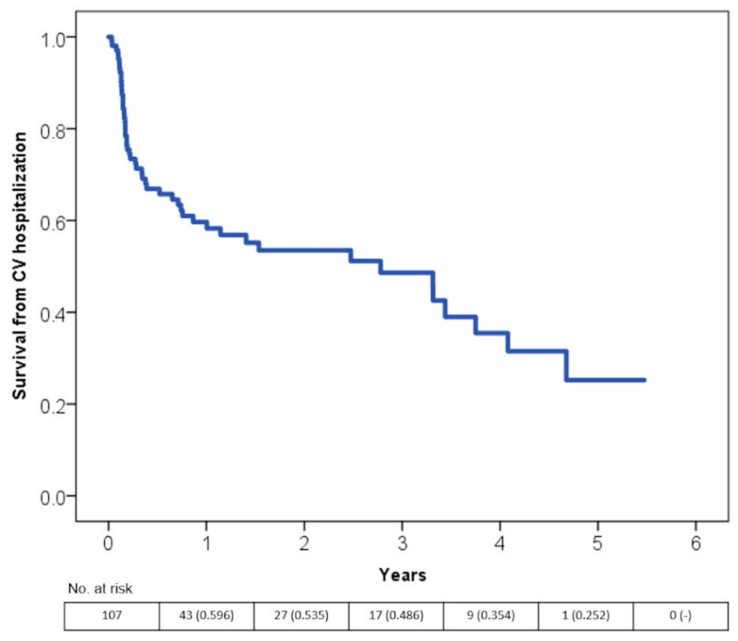
Time free from hospitalization for cardiovascular reasons according to the Kaplan-Meier curves.

**Table 1 jcm-08-01411-t001:** Baseline characteristics, also divided by the incidence of tachycardiomyopathy recurrence and cardiovascular-related hospitalization.

Variable	Total Population (*n* = 107)	No Recurrence (*n* = 90)	Recurrence (*n* = 17)	*p* Value	No CV Hospitalization (*n* = 56)	CV Hospitalization (*n* = 51)	*p* Value
Male gender	68 (64%)	58 (64%)	10 (59%)	0.659	32 (57%)	36 (71%)	0.149
Age (years)	66.7 ± 14.5	66.9 ± 15.1	66.0 ± 11.3	0.816	68.8 ± 15.4	64.4 ± 13.2	0.117
BMI (Kg/m^2^)	28.6 ± 5.3	28.5 ± 5.3	29.1 ± 5.3	0.692	28.4 ± 5.7	28.7 ± 5.0	0.655
Hypertension	68 (64%)	57 (63%)	11 (65%)	0.911	39 (67%)	29 (57%)	0.170
Diabetes	14 (13%)	14 (15%)	0 (0%)	0.081	7 (12%)	7 (14%)	0.851
Dyslipidaemia	36 (34%)	29 (32%)	7 (41%)	0.474	19 (34%)	17 (33%)	0.948
CKD	22 (21%)	20 (22%)	2 (12%)	0.328	12 (21%)	10 (20%)	0.816
COPD	12 (11%)	11 (12%)	1 (6%)	0.447	9 (16%)	3 (6%)	0.095
OSAS	6 (6%)	2 (2%)	4 (24%)	0.006	1 (2%)	5 (10%)	0.100
Hyperthyroidism	6 (6%)	5 (5%)	1 (6%)	0.273	5 (9%)	1 (2%)	0.118
Hypothyroidism	7 (7%)	4 (4%)	3 (17%)	0.043	3 (5%)	4 (8%)	0.707
AF as trigger	83 (77%)	67 (74%)	16 (94%)	0.075	43 (77%)	40 (78%)	0.838
On admission:							
NYHA class II	18 (17%)	15 (17%)	3 (18%)	0.730	9 (16%)	9 (18%)	0.970
NYHA class III	62 (58%)	51 (57%)	11 (65%)		33 (59%)	29 (60%)	
NYHA class IV	27 (25%)	24 (27%)	3 (17%)		14 (25%)	13 (25%)	
Heart rate (bpm)	126.5 ± 28.9	127.0 ± 30.8	124.2 ± 17.5	0.714	127.2 ± 23.1	125.9 ± 34.2	0.818
BNP (pg/mL)	575 (312–786)	541 (293–771)	781 (655–1247)	0.012	547 (364–765)	624 (270–851)	0.413
Troponin I (ng/mL)	0.02 (0.01–0.06)	0.02 (0.01–0.06)	0.03 (0.01–0.07)	0.694	0.02 (0.01–0.06)	0.03 (0.01–0.08)	0.146
LVEF (%)	32.9 ± 9.7	32.9 ± 9.4	32.6 ± 8.7	0.918	34.2 ± 7.9	31.4 ± 10.4	0.129
iLAV (mL/m^2^)	50.15 ± 14.5	48.2 ± 14.0	58.7 ± 13.9	0.037	49.5 ± 11.8	51.0 ± 17.5	0.717
At discharge:							
Heart rate (bpm)	71.0 ± 15.0	69.5 ± 14.8	78.2 ± 14.4	0.029	68.4 ± 13.5	73.9 ± 16.2	0.067
BNP (pg/mL)	257 (124–511)	244 (123–429)	307 (169–670)	0.141	165 (89–252)	354 (249–551)	0.02
Troponin I (ng/mL)	0.03 (0.01–0.04)	0.05 (0.01–0.06)	0.02 (0.01–0.04)	0.99	0.02 (0.01–0.05)	0.03 (0.02–0.03)	0.99
LVEF (%)	41.0 ± 11.8	41.8 ± 12.1	36.0 ± 8.8	0.179	43.5 ± 9.8	37.5 ± 13.6	0.047
NYHA class I	27 (26%)	24 (27%)	3 (17%)	0.655	16 (29%)	11 (22%)	0.431
NYHA class II	74 (70%)	61 (69%)	13 (76%)		39 (67%)	35 (71%)	
NYHA class III	4 (4%)	3 (3%)	1 (6%)		1 (2%)	3 (6%)	

AF: atrial fibrillation; BMI: body mass index; BNP: brain natriuretic peptide; CKD: chronic kidney disease; COPD: chronic obstructive pulmonary disease; CV: cardiovascular; iLAV: indexed left atrial volume; LVEF: left ventricular ejection fraction; NYHA: New York Heart Association; OSAS: obstructive sleep apnoea.

**Table 2 jcm-08-01411-t002:** Treatment strategies at discharge.

Variable	Total Population (*n* = 107)	Mean TIS *
ACE-Inhibitors	59 (55%)	0.40 ± 0.26
ARBs	34 (32%)	0.46 ± 0.36
Beta-blockers	98 (92%)	0.54 ± 0.24
MRAs	86 (80%)	0.50 ± 0.24
Loop diuretics	92 (86%)	49.73 ± 36.55 **
Ivabradine	2 (2%)	0.50
Flecainide	4 (4%)	0.50
Amiodarone	57 (53%)	0.97 ± 0.09
Digoxin	11 (10%)	0.45 ± 0.22
CCBs	12 (11%)	0.75 ± 0.23
Pharmacological cardioversion	11 (10%)	
Electrical cardioversion	68 (64%)	
Catheter ablation	18 (17%)	
Successful rhythm control	67 (63%)	
WCD	10 (9%)	

* The mean therapeutic index was calculated only in those patients who were administered the drug at least until discharge. ** For loop diuretics we considered the total dose per day as equivalents of furosemide. ACE-I: angiotensin converting enzyme inhibitor; ARB: angiotensin II receptor blocker; CCB: calcium-channel blocker; MRA: mineralocorticoid receptor antagonist; WCD: wearable cardioverter-defibrillator.

**Table 3 jcm-08-01411-t003:** Multivariable Cox-proportional hazard model for tachycardiomyopathy recurrence.

Variable	HR	95% CI Lower Bound	95% CI Lower Bound	*p* Value
OSAS	5.88	1.38	17.29	0.045
BNP at admission (for each pg/mL)	1.01	1.01	1.03	0.014
Heart rate at discharge (for each bpm)	1.05	1.01	1.10	0.029

**Table 4 jcm-08-01411-t004:** Multivariable Cox-proportional hazard model for cardiovascular hospitalization.

Variable	HR	95% CI Lower Bound	95% CI Lower Bound	*p* Value
LVEF at discharge (for each%)	0.96	0.93	0.99	0.020
Heart rate at discharge (for each bpm)	1.02	1.01	1.04	0.032

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
