# Peer review of "Tachycardiomyopathy in Patients without Underlying Structural Heart Disease"

_jcm, 2019, doi:10.3390/jcm8091411_

Round 1

Reviewer 1 Report

The manuscript by Stronati et al. is about the risk factors affecting the recurrence of tachycardia-induced cardiomyopathy. The manuscript is interesting however I think it would benefit from a more detailed introduction, especially considering the importance of this underestimated case of cardiomyopathy.

Major:  P10 Figure 2 was not included to the manuscript.

Minor:

P.1 Line 20: Affiliation after “and”. Please check the text.

P.1. Line 24: Please check the text “Correspondence: Contributed equally….. “

P2 Line 37:  The abbreviation “HR” should be defined.

Abbreviation HR is used for Hazard Ratio and Heart Rate.

I suggest not to use the abbreviation for “heart rate” and write it in full (in tables and figures and in the overall text,).

P4. Line 98. Undefined abbreviation EF

P10. Undefined abbreviation TIA. Please do not overuse abbreviations.

P10. Was the death of 58 yo patient related to the cardiovascular disease?

P11 Line 16, P12 Line 42: “from Table 1” Do authors mean to say “Table 2”

P11. Figure 3. Figure parts are not labelled (a, b).

P11 Lines 11-19, P12 Lines 37-45. Would it be better to describe it in Methods (Statistical Analysis) rather than here?

P13. Line 66. Abbreviation CPAP is undefined. Write it in full.

Author Response

Review #1

The manuscript by Stronati et al. is about the risk factors affecting the recurrence of tachycardia-induced cardiomyopathy. The manuscript is interesting however I think it would benefit from a more detailed introduction, especially considering the importance of this underestimated case of cardiomyopathy.

We thank reviewer #1 for his comments. We have improved the introduction according to his request.

Major: P10 Figure 2 was not included to the manuscript.

We are sorry for the inconvenience. We added Figure 2 in the text.

Minor:

P.1 Line 20: Affiliation after “and”. Please check the text.

We confirm that the affiliation is correct and correctly spelt.

P.1. Line 24: Please check the text “Correspondence: Contributed equally….. “

We corrected the phrase as requested.

P2 Line 37: The abbreviation “HR” should be defined. Abbreviation HR is used for Hazard Ratio and Heart Rate. I suggest not to use the abbreviation for “heart rate” and write it in full (in tables and figures and in the overall text,).

We amended the paper as requested by reviewer #1, spelling our “heart rate” throughout the text and defining “HR” as “hazard ratio” the first time encountered both in the abstract and in the main text.

P4. Line 98. Undefined abbreviation EF

We defined LVEF as left ventricular ejection fraction and is now consistent throughout the text.

P10. Undefined abbreviation TIA. Please do not overuse abbreviations.

We spelt out “transient ischemic attack” throughout the text in order to reduce the number of abbreviations.

P10. Was the death of 58 yo patient related to the cardiovascular disease?

We added the requested info regarding the death of our patient as requested. Having had an out-of-hospital cardiac arrest, we do not have all the details.

P11 Line 16, P12 Line 42: “from Table 1” Do authors mean to say “Table 2”

Thank you for pointing out the mistake, which was now corrected.

P11. Figure 3. Figure parts are not labelled (a, b).

We labelled Figure 3 as requested. Subfigures are now in the correct order.

P11 Lines 11-19, P12 Lines 37-45. Would it be better to describe it in Methods (Statistical Analysis) rather than here?

Thank you for your comment. We preferred to add every single covariate in the table legends in order to help the reader understand which variables were tested without looking back and forth throughout the manuscript. The list of covariates can be, however, moved to the statistical analysis section if deemed necessary.

P13. Line 66. Abbreviation CPAP is undefined. Write it in full.

We have spelt out CPAP as requested.

Reviewer 2 Report

This is an interesting analysis of TIC, which adds to the limited literature on this topic.

The key limitation here is that the definition used for TIC seems quite mild, given an improvement of LVEF of 5% is fairly small and with observer variability, particularly in the setting of tachycardia. Given this was first presentation, these changes could be due to starting HF medications. Could the authors address this. Perhaps they could provide details on the patients who truly fully recovered to a normal EF (say >50%) - was there any recurrence/hospitalisation in this group?

Is it possible that improvements in LV function were simply as a result of HF treatment rather than rate control?

Who measured LVEF? Was it one person or two and were they blinded? This is important given the difficulties in echo LVEF assessment in context of tachycardia. A core lab would have been ideal, accepting likely to not be feasible, this should still be in the limitations.

How many patients were in sinus rhythm at discharge?

Was there any difference between those in whom a rate as opposed to rhythm control strategy was used?

Figure 3 graphs are presented the wrong way round.

Could the authors present the heart rate and LVEF over time separately for the recurrence group? It would be interesting to know whether they matched the non-recurrent group, for example showing less EF recovery in the interim period before decompensation.

Was there any difference in medication use between those who had recurrence and those who didn't?

Was treatment kept the same over the duration of follow-up or did some people stop? Did some people have intensification of therapy? This might influence events over time.

The group that had a further CV hopsitalisation or recurrence seemed sicker, with higher heart rates, LA volume, BNP etc. I do wonder if there are two separate groups here, including one without TIC.

Author Response

Review #2

This is an interesting analysis of TIC, which adds to the limited literature on this topic.

The key limitation here is that the definition used for TIC seems quite mild, given an improvement of LVEF of 5% is fairly small and with observer variability, particularly in the setting of tachycardia. Given this was first presentation, these changes could be due to starting HF medications. Could the authors address this. Perhaps they could provide details on the patients who truly fully recovered to a normal EF (say >50%) - was there any recurrence/hospitalisation in this group?

We thank reviewer #2 for underlying this important point. We agree that an improvement of >5% could seem rather small, but unfortunately, there is no consensus on any sort of cut-off for TCM. We chose this cut-off because we thought that, after ruling out all structural heart diseases, a patient undergoing an improvement of both NYHA class and EF could be considered as having TCM, being the arrhythmia the only remaining cause of his/her condition. A higher cut-off could have ruled out many TCM that just had not time to recover completely because of recurrence, without giving another plausible explanation for the HF episode. We, however, agree with reviewer #2 on the fact that this cut-off is somewhat arbitrary and have included this point in the limitations section. Regarding HF treatment, please note that none of the HF drug classes is considered as an independent predictor of recurrence or CV hospitalization in the multivariable Cox proportional model (Table 3 and 4), strengthening the idea that heart rate control is the main driver of recovery, rather than HF medications. Please also consider that both patients with recurrences and without recurrences reached an EF ≥50% at 12-months (please see the new figure 3b, modified according to reviewer #2’ comments). Also, comparing the mean TIS in patients with/without recurrences and with/without CV hospitalizations held no significant results. We updated the methods, results and discussion sections accordingly.

Is it possible that improvements in LV function were simply as a result of HF treatment rather than rate control?

We agree that HF treatment could be considered as a confounder in the association between TCM and EF improvement. However, heart rate at discharge was a predictor of both recurrence and cardiovascular hospitalization, independently of any kind of pharmacological treatment at discharge (Table 3 and 4). As the criteria for recurrence are the same as for the first TCM, we can hypothesise that, in our patients, heart rate is what matters and the association with EF worsening could be considered independent of HF treatment. We added this point in the limitation section.

Who measured LVEF? Was it one person or two and were they blinded? This is important given the difficulties in echo LVEF assessment in context of tachycardia. A core lab would have been ideal, accepting likely to not be feasible, this should still be in the limitations.

All echocardiograms were performed for clinical reasons and stored into our image server. Regarding the present analysis, all echo exams were reviewed by two authors (G.S. and F.G.), who were responsible for the off-line analysis and collected all measurements. Being a monocentric study, we thought that a core lab would not be necessary, as the two authors reviewed all the clips and resolved all conflicts by consensus. We also added inter-operator and intra-operator coefficients of variations, which were way lower than the chosen 5% cut-off for LVEF. We amended the methods section in order to add this part as requested.

How many patients were in sinus rhythm at discharge?

From Table 2, 67 out of 107 (63%) were successfully restored to sinus rhythm.

Was there any difference between those in whom a rate as opposed to rhythm control strategy was used?

We thank reviewer #2 for helping us in improving this part of the paper. We added a dummy variable “rhythm control vs. rate control” in the multivariable Cox-proportional hazard models. Adding such a variable to our statistical models does not modify any of the results. We modified Table 3 and 4 accordingly. Please also take a look at Supplementary Table 2 and 3, showing all the Cox models in detail.

Figure 3 graphs are presented the wrong way round.

Subfigures are now in the correct order.

Could the authors present the heart rate and LVEF over time separately for the recurrence group? It would be interesting to know whether they matched the non-recurrent group, for example showing less EF recovery in the interim period before decompensation.

We thank reviewer #2 for this suggestion, as we think that it enabled us to vastly improve our paper. We modified Figure 3 as requested in order to show LVEF and heart rate values for both recurrence and no recurrence groups. We also performed a T-test for independent sample in order to test statistical significance for each time point. As the reviewer can see, EF is not different between the two groups in any of the timepoint considered, and is absolutely the same at 1-year, being above 50% for both groups. On the other hand, heart rate is significantly higher in patients with a future recurrence at discharge, after 3 months and after 1 year too, consistently with the other models already presented in the original version of our paper.

Was there any difference in medication use between those who had recurrence and those who didn't?

As already mentioned before, the only independent predictors of recurrence were OSAS, BNP at admission and heart rate at discharge. Nonetheless, we performed an ANOVA comparing the TIS for each treatment class between patients with and without recurrences and found no significant differences. We added a sentence in the results section to address this result.

Was treatment kept the same over the duration of follow-up or did some people stop? Did some people have intensification of therapy? This might influence events over time.

We thank the reviewer for the question. Treatment was changed according to the guidelines when and as needed. However, it is almost impossible to register and report all the treatment changes in statistical analysis. We addressed this limitation in the proper section.

The group that had a further CV hopsitalisation or recurrence seemed sicker, with higher heart rates, LA volume, BNP etc. I do wonder if there are two separate groups here, including one without TIC.

We thank review #2 for raising this important point. As already mentioned in Table 4, the only independent predictors of CV hospitalization were LVEF and heart rate at discharge, with LA volume, BNP levels and other comorbidities not reaching statistical significance. Furthermore, as described in the methods section, we thoroughly excluded patients with any structural heart disease or with no evidence of improvement after discharge. While being, of course, a retrospective study with small sample size and many limitations, our current data do not lead us to believe that patients with CV hospitalizations can represent a distinct group, less more so they can be considered as not having TCM.

Reviewer 3 Report

The manuscript submitted by Stronati et al. is well written and the topic of high interest. The authors identified possible predictors of recurrence and long-term morbidity in patients suffering from pure tachycardiomyopathy. Remarkably, about 1200 patients were screened and most of the inclusion and exclusion criteria are well defined.

However, some points need revision and have to be further clarified.

Major points:
1) Despite listing the inclusion and exclusion criteria it remains unclear if all patients with CAD were excluded or only those with diagnosis of ischemia.
2) The chosen method to determine recovery from TCM by improvement of at least one NYHA class and five points of EF appears questionable. In the manuscript of the first cited author (Martin et al.) I am not able to detect this definition. In the manuscript of Jeong et al. all TCM patients became asymptomatic and deltaEF improved at least by 15 points. The authors should explain and justify the chosen definition of recovery. The cited manuscripts do not seem to support this definition.
3) 17 patients experienced recurrence of tachycardiomyopathy. The authors defined recurrence as “a new episode of acute HF with reduced ejection fraction along with …”. This explanation is not sufficient. If recovery is defined as an increase of EF by 5 points, is a decrease of 5 points with concomitant arrhythmia similar to recurrence? Please explain.
4) ACE-Inhibitors and ARBs taken together 87% (55% + 32%) of patients received drugs recommended for treatment of symptomatic HFrEF but 13% did not. Quite a number of patients did not seem to have received the most essential drug for HFrEF treatment at discharge. Please comment on that issue.
5) The discussion does not appear focused.
a. The authors state that about 10% of hospitalizations for acute HF might be TCM associated and that early recognition of the triggering arrhythmia can lead to treatment strategies. Is there a feasible approach? Can the authors give an outlook how this could be realized?
b. It is mentioned that TCM recurrence is higher between the fifths and sixth year after diagnosis and the authors speculate that reduction of patients’ adherence to treatment might be responsible. Therefore, the chosen treatment in this groups is of interest and should be included. Additionally, the most common observed arrhythmia was atrial fibrillation which is known to have a progressive course. Which arrhythmias were documented in patients with TCM recurrence? Could it be that paroxysmal AF advanced to persistent AF, thereby causing recurrence of TCM?
c. It is emphasized that the heart rate at discharge is associated with an increased risk of TCM recurrence which warrants discussion. However, this finding is not really discussed in the context of TCM treatment. Moreover, the authors state that this finding is in contrast to evidence in AF patients. Why is that? Is there a possible explanation? Please discuss.
d. The authors discuss their finding that increased BNP values at admission are associated which TCM recurrence. To do so, they cite a study which identified BNP to be of value in TCM identification, which does not support this statement. Furthermore, they claim that their finding supports the hypothesis that TCM patients benefit from continuation of HF treatment even after normalization of LVEF, as structural cardiac abnormalities persisting after recovery cannot be ruled out. Why? This assertion appears exaggerated. Is there an association between BNP-levels at baseline of hospitalization and sufficient long-time HF treatment with normal EF or TCM? Please add a reference supporting your claim.
e. It is discussed that TCM recovery might represent a transition to a subclinical form of heart remodeling rather than improvement of LVEF. This statement does not seem to be associated to any of the studies aims and results and should be excluded.
f. The authors refer to relatively low BNP levels as a limitation. They explain that slightly elevated BNP levels, which are still significantly higher than the cut-offs in guidelines, give a clue that patients with dyspnea as sole manifestation of AF and no overt HF were correctly excluded. I do not understand this reasoning. AF patients without heart failure also have elevated BNP levels above the cut-offs mentioned in guidelines (e.g. Kotecha et al., JACC, 2016).

Please focus the discussion.

Minor points
1) In the abstract and materials and methods part the authors refer to the study as “prospective and observational” and in the limitations part as “retrospective and observational”. Please comment on this issue.
2) Is the cause of death in the male patient known or not know? The secondary endpoint “major adverse cardiovascular events” including non-fatal stroke, non-fatal myocardial infarction and cardiovascular death is not mentioned in the results part anymore. Two non-fatal strokes and no cardiovascular death are described. Did any patient experience myocardial infarction? Maybe a summary of events in the supplementary material could be helpful.
3) The authors explicitly mention that no Sotalol and Flecainide were used. Other antiarrhythmic drugs like dronedarone and vernakalant are not mentioned. Is the information which drugs were not used essential for the results? It might be better to report which AAD class was used.
4) It appears that Figure 2 is not included in the manuscript.
5) The results of the univariate model and not significant parameters of the multivariate model should be included in the supplementary material. Were some of them significant by trend?
6) It would be interesting how the 17 patients suffering from TCM recurrence were treated, if there was a change in the treatment strategy and if the chosen strategy has been successful. Can this be added?
7) Figure 3 displays mean values of left ventricular EF and heart rate on admission, at discharge, during follow-up and at recurrence. It is not well explained which collective these results refer to. Supposedly, the authors refer to the 17 patients with TCM recurrence. If so, the values for heart rate and LVEF on admission and discharge do not seem to match the corresponding values listed in table 1. This appears illogical to the reader.

In summary, there are some major and minor points of criticism regarding the submitted manuscript. Especially the discussion does not appear focused on the authors results and needs extensive revision. Additionally, a major weak point is the definition of TCM recovery and recurrence, which is the foundation of the presented study.

Therefore, the manuscript does not seem to be fit for publication in the Journal of Clinical Medicine in its current form. I would support re-submission of this manuscript in its revised form.

Author Response

Reviewer #3

The manuscript submitted by Stronati et al. is well written and the topic of high interest. The authors identified possible predictors of recurrence and long-term morbidity in patients suffering from pure tachycardiomyopathy. Remarkably, about 1200 patients were screened and most of the inclusion and exclusion criteria are well defined. However, some points need revision and have to be further clarified.

We thank reviewer #3 for his/her important effort in strengthening our paper.

Major points:

1) Despite listing the inclusion and exclusion criteria it remains unclear if all patients with CAD were excluded or only those with diagnosis of ischemia.

As written in the methods section, ischemic heart disease was defined as a previous history of revascularization, or evidence of significant coronary obstruction at coronary angiography performed during hospitalization. Patients with non-significant CAD and no clinical of instrumental signs of ischemia were not excluded. This has been added in the methods section.

2) The chosen method to determine recovery from TCM by improvement of at least one NYHA class and five points of EF appears questionable. In the manuscript of the first cited author (Martin et al.) I am not able to detect this definition. In the manuscript of Jeong et al. all TCM patients became asymptomatic and deltaEF improved at least by 15 points. The authors should explain and justify the chosen definition of recovery. The cited manuscripts do not seem to support this definition.

We agree with reviewer #3 that our definition of TCM is probably the hottest point of the present manuscript as it is, as the reviewer implied, an arbitrary value. Unfortunately, in current literature there is absolutely no consensus on what a “good” cut-off value should be, neither if there should be an EF cut-off at all. The most cited reviews on the subjects do not provide any kind of cut-off (Martin, C.A. Heart 2017, 103, 1543–1552: Simantirakis; Europace 2012, 14, 466–473; Gopinathannair, R; J. Am. Coll. Cardiol. 2015), and this could be, at least in part, one of the reasons behind the lack of evidence on a so frequent cause of ventricular dysfunction. The only available cut-off is the one already mentioned by the reviewer, i.e. an increase of 15% and complete symptoms recovery (Jeong, Y. Clin. Cardiol. 2008, 31, 172–178). Unfortunately, this study was made on 21 patients (less than one-fifth of our population) and their cut-off was as arbitrary as ours. Moreover, such a high cut-off in EF, while giving a false sense of security in order to cut out “false positives” has the drawback to exclude many patients who recovered more slowly or in which the recovery was hampered by early arrhythmic recurrence. As reviewer #3 can see from the new figure 3b, all our patients reached 50% EF at one-year, so the possibility to select “false” TCM seems unlikely, at least from our limited data. So, to cite the reviewer #3’s words, we agree that “the cited manuscripts do not seem to support this [ours] definition” but it must be noted that current literature does not currently support any definition either. Also, we do not feel that another arbitrary definition on a smaller sample (i.e. Jeong’s cut-off) and many other drawbacks should be used, with the sole reason that is already published. We, of course, agree with reviewer #3 that this could be a major limitation of our study, and have extensively addressed it in the limitations section of our revised version.

3) 17 patients experienced recurrence of tachycardiomyopathy. The authors defined recurrence as “a new episode of acute HF with reduced ejection fraction along with …”. This explanation is not sufficient. If recovery is defined as an increase of EF by 5 points, is a decrease of 5 points with concomitant arrhythmia similar to recurrence? Please explain.

We thank reviewer #3 for the opportunity to clarify this issue, but this is not actually correct. In the original version of our manuscript, we state that “Primary endpoint was recurrence of TCM, defined as a new episode of acute HF with reduced ejection fraction along with evidence of atrial or ventricular arrhythmia, occurring after clinical and echocardiographic recovery of the original episode.” While wording can be improved (and has been in the revised version of our manuscript), we did not mention a decrease of 5% anywhere in the text. We updated the definition of recurrence in the methods section as follows, in order to be more precise: “a new episode of acute HF with reduced ejection fraction along with evidence of atrial or ventricular arrhythmia, occurring after complete clinical (no HF symptoms) and echocardiographic (LVEF ≥50%) recovery of the original episode.”

4) ACE-Inhibitors and ARBs taken together 87% (55% + 32%) of patients received drugs recommended for treatment of symptomatic HFrEF but 13% did not. Quite a number of patients did not seem to have received the most essential drug for HFrEF treatment at discharge. Please comment on that issue.

The 13% of patients that did not receive the most essential drug for HFrEF at discharge because their EF recovered quickly during hospitalization. Therefore, according to the current European guidelines, such treatment was not recommended anymore. Please also look at the new Figure 3b for mean LVEF values at discharge.

5) The discussion does not appear focused.

The authors state that about 10% of hospitalizations for acute HF might be TCM associated and that early recognition of the triggering arrhythmia can lead to treatment strategies. Is there a feasible approach? Can the authors give an outlook how this could be realized?

We thank reviewer #3 for pointing out this interesting aspect. We added a paragraph in the discussion section (page 14) in which we suggest how early recognition of a triggering arrhythmia could help to optimize treatment, and we suggest a feasible management strategy.

It is mentioned that TCM recurrence is higher between the fifths and sixth year after diagnosis and the authors speculate that reduction of patients’ adherence to treatment might be responsible. Therefore, the chosen treatment in this groups is of interest and should be included. Additionally, the most common observed arrhythmia was atrial fibrillation which is known to have a progressive course. Which arrhythmias were documented in patients with TCM recurrence? Could it be that paroxysmal AF advanced to persistent AF, thereby causing recurrence of TCM?

We included treatment strategies for recurrence in the results section (please also see reply to minor point #6). As mentioned in the new and revised results section, 15 out of 17 patients with TCM recurrence experienced AF as the triggering arrhythmia, so the association between the progression of AF and recurrence makes sense. We edited the discussion in order to elaborate on this important suggestion (page 15).

It is emphasized that the heart rate at discharge is associated with an increased risk of TCM recurrence which warrants discussion. However, this finding is not really discussed in the context of TCM treatment. Moreover, the authors state that this finding is in contrast to evidence in AF patients. Why is that? Is there a possible explanation? Please discuss.

We thank reviewer #3 for giving us the opportunity to expand on this topic. We improved the discussion by adding a segment to the discussion along with some relevant references. The present paragraph has been added in the discussion section: “Reasons for such a striking difference can be found in the different pathophysiological mechanisms. In the RACE II trial, AF patients with severe HF or with recent decompensation were excluded, thus leaving only patients without HF or with stable mild symptoms for at least three months.[20] In this setting, it has already been demonstrated that the actual benefit from heart rate reduction and sinus rhythm restoration could be counterbalanced by the increased likelihood of adverse effects due to anti-arrhythmic drugs[21] and, therefore, pushing too hard on heart rate reduction could produce no further clinical benefits. On the other hand, it is well known that heart rate is a risk factor in patients with HF, even when sinus rhythm is present. Dysfunctional myocardium is energetically depleted and myocardial exerted force is negatively associated with the rate of contraction.[22] In an HF setting, such as the one of TCM occurrence, reducing heart rate improves contractility, extends coronary diastolic filling time, reduces energy expenditure and improves cardiac output.[23] Moreover, the benefits of a reduced heart rate are consistent over years, due to positive modifications of the extracellular matrix and myocytes properties,[24] resulting in a reduced risk of cardiovascular events and HF recurrences over a long follow-up.”

The authors discuss their finding that increased BNP values at admission are associated which TCM recurrence. To do so, they cite a study which identified BNP to be of value in TCM identification, which does not support this statement. Furthermore, they claim that their finding supports the hypothesis that TCM patients benefit from continuation of HF treatment even after normalization of LVEF, as structural cardiac abnormalities persisting after recovery cannot be ruled out. Why? This assertion appears exaggerated. Is there an association between BNP-levels at baseline of hospitalization and sufficient long-time HF treatment with normal EF or TCM? Please add a reference supporting your claim.

As reviewer #3 perfectly knows, there are no references confirming the association between BNP levels at admission and complete EF recovery in TCM, as the present study represents the first and largest study providing that sort of evidence. We were probably misinterpreted as we were making no “claims” nor “assertions” whatsoever, but only giving potential hypotheses which could stem from our results (hence, for example, the use of conditional tenses throughout most of the discussion). Nonetheless, we edited the paragraph in order to remove any hint of false certainty from our sentences.

It is discussed that TCM recovery might represent a transition to a subclinical form of heart remodeling rather than improvement of LVEF. This statement does not seem to be associated to any of the studies aims and results and should be excluded.

We deleted the statement as requested.

The authors refer to relatively low BNP levels as a limitation. They explain that slightly elevated BNP levels, which are still significantly higher than the cut-offs in guidelines, give a clue that patients with dyspnea as sole manifestation of AF and no overt HF were correctly excluded. I do not understand this reasoning. AF patients without heart failure also have elevated BNP levels above the cut-offs mentioned in guidelines (e.g. Kotecha et al., JACC, 2016).

We removed the paragraph on BNP levels from the limitations section as, as correctly pointed out, was potentially misleading.

Minor points

1) In the abstract and materials and methods part the authors refer to the study as “prospective and observational” and in the limitations part as “retrospective and observational”. Please comment on this issue.

We thank reviewer #3 for pointing out this typo. The study is prospective, and the error in the limitations part has been amended.

2) Is the cause of death in the male patient known or not know?

We have incomplete data regarding the only death as it was due to an out-of-hospital cardiac arrest. However, a diagnosis of idiopathic ventricular fibrillation was made by the paramedics, and we added all the relevant data in the results section of the manuscript.

The secondary endpoint “major adverse cardiovascular events” including non-fatal stroke, non-fatal myocardial infarction and cardiovascular death is not mentioned in the results part anymore. Two non-fatal strokes and no cardiovascular death are described. Did any patient experience myocardial infarction? Maybe a summary of events in the supplementary material could be helpful.

We agree with reviewer #3 that cardiovascular events deserve more space in the results section. We amended results adding the following: “Among major adverse cardiovascular events, two patients suffered from thromboembolic events (1.8%) and one died for cardiovascular causes (0.9%). No non-fatal myocardial infarctions were reported. The annual incidence of recurrence was 8.4% per year, 0.9% per year for thromboembolic events and 0.4% per year for cardiovascular mortality.”

3) The authors explicitly mention that no Sotalol and Flecainide were used. Other antiarrhythmic drugs like dronedarone and vernakalant are not mentioned. Is the information which drugs were not used essential for the results? It might be better to report which AAD class was used.

We edited out of the tables all the drugs which were not used at all.

4) It appears that Figure 2 is not included in the manuscript.

We apologize for the inconvenience. Figure 2 has been now added and referenced correctly in the text.

5) The results of the univariate model and not significant parameters of the multivariate model should be included in the supplementary material. Were some of them significant by trend?

We provided the complete and full univariable and multivariable models as supplementary tables 2 and 3. However, we prefer to refrain from commenting any “trend”, as the notion of trend does not exist from a statistical point of view, and what can be said in these cases is only that no significant differences have been found. The low sample size could be responsible for the lack of statistical significance, with no evidence that adding more patients could “improve the trend” in one way or another.

6) It would be interesting how the 17 patients suffering from TCM recurrence were treated, if there was a change in the treatment strategy and if the chosen strategy has been successful. Can this be added?

We included treatment strategies for recurrence in the results section. In brief, out of the 17 patients experiencing recurrences, 14 underwent catheter ablation of AF (12 patients) or atrial flutter (2 cases). Patients undergoing catheter ablation of atrial flutter experienced no further TCM recurrences, while five patients presented a second TCM recurrence after the procedure. Three patients with AF refused to consent to catheter ablation and were therefore shifted to a rate-control strategy, with one patient experiencing no further recurrences, one experiencing a second recurrence and another patient experiencing three more recurrences over follow-up.

7) Figure 3 displays mean values of left ventricular EF and heart rate on admission, at discharge, during follow-up and at recurrence. It is not well explained which collective these results refer to. Supposedly, the authors refer to the 17 patients with TCM recurrence. If so, the values for heart rate and LVEF on admission and discharge do not seem to match the corresponding values listed in table 1. This appears illogical to the reader.

We provided a new figure 3b in which heart rate and LVEF were split according to the presence or absence of recurrence. Of course, mean heart rate and LVEF values in the “recurrence” column consider only patients who actually had a recurrence. Now it has been clarified in the graphs. We also checked that all values were consistent between Figure 3 and Table 1.

Round 2

Reviewer 1 Report

I am satisfied with the answers and corrections.

Thank you.

Reviewer 2 Report

The authors have done a good job and have addressed all of my comments adequately. The paper is much improved.

Reviewer 3 Report

The scientific quality of the present manuscript has significantly improved during the the review process. However, i still have concerns about the used definition of pump function recovery in the study group. Hence, i appreciate that this issue is now discussed appropriately in the present form of the manuscript.